# How Can We Reduce Dental Fear in Children? The Importance of the First Dental Visit

**DOI:** 10.3390/children8121167

**Published:** 2021-12-09

**Authors:** María Carrillo-Díaz, Blanca Carmen Migueláñez-Medrán, Carolina Nieto-Moraleda, Martín Romero-Maroto, María José González-Olmo

**Affiliations:** 1Department of Paediatric Dentistry, Rey Juan Carlos University, 28922 Alcorcón, Spain; 2Department of Nursing and Dentistry, Rey Juan Carlos University, 28922 Alcorcón, Spain; blancac.miguelanez@urjc.es; 3Department of Orthodontics, Rey Juan Carlos University, 28922 Alcorcón, Spain; carolina.nieto@urjc.es (C.N.-M.); martin.romero@urjc.es (M.R.-M.); mariajose.gonzalez@urjc.es (M.J.G.-O.)

**Keywords:** dental fear, dental anxiety, preventive dentistry, clinic visit, children

## Abstract

Dental fear is a common problem amongst children. It can affect children’s psychological well-being, quality of life, and oral and systemic health. The aim of this study was to identify whether the patients’ age at which visits to the paediatric dentist begin as well as the periodicity of these visits are factors that can prevent dental fear. This observational transversal study was conducted on 575 school children (average age 6.85 ± 0.78) and their mother/father/guardian. Parents completed a survey on the characteristics of dental visits and the child completed the index of dental anxiety and fear (IDAF-4C) to assess dental fear. The correlation between dental fear and age at first visit (r = −0.36 *p* < 0.01) and dental fear and frequency of visit (r = −0.65 *p* < 0.01) were statistically significant. The regression analysis performed showed that both variables predicted 44.4% of the dental fear in the child. In conclusion, the age of initiation to the paediatric dentist (before 2 years) and the periodic revisions (every 6 months or every year) could protect the child from dental fear.

## 1. Introduction

Fear acts, basically, as a system that alerts people to danger [1]. The oral region is very sensitive intervenes in important processes, such as speech and chewing. This contributes to the difficulties that dentists often face in their clinical practices, including the dental fear that affects children [1,2]. Estimations obtained in different studies show elevated variability in the prevalence of dental fear. It has been calculated that dental anxiety is a condition that affects approximately 9% of children and adolescent in United States and Canada, Australia, and Europe [3]. Similarly, Grisolia BM et al. summarize that dental anxiety is a frequent problem in pediatric population worldwide, particularly in school and preschool children than in adolescents [4]. However, a 26% prevalence was observed in Italian kids from 6 to 10 years old [5] while only 4.9% prevalence was observed in Spanish population from 7 to 12 years old [6]. In addition, a 16.1% of Australian children older than 5 years were reported to present high levels of dental fear [7]. Differences on the measurements, the methods, or criteria as well as changes on the sample or in the definition of dental fear itself may account for the observed variability [8].

This fear generally links to three important consequences for children: they may use dental services less than necessary, they may not be able to receive certain necessary dental treatments, and they may not be cooperative in the professional practice [9,10,11,12,13]. In other words, dental fear can be a source of serious health problems; it can affect children’s psychological well-being, quality of life, and oral and systemic health.

In general, dentist visits and dental fear are easily linked. Overcoming this fear is important because children who attend the dentist regularly are more likely to have positive and safe dental practices, which can facilitate progressive familiarization with dental care. This way, future anxiety can be prevented at an early age [14,15]. In other words, regular dentist visits that are not painful or adverse for the child act as a preventive measure against dental fear [15].

The classical conditioning mechanism seems to operate on dental fear. A report from Townend E et al. [15] indicates that, in many cases (they included sixty children from two age groups: 7–10 years, 11–14 years) maladaptive fear and avoidance behaviors arise after unpleasant experiences when undergoing various dental procedures [16]. Specifically, Öst and Husgahl [17] found that 68.6% of dental phobias were due to patients’ traumatic experiences during dental treatment. From a behavioral perspective, fear responses have been considered learnt responses, and patients’ traumatic or unpleasant experiences are considered classic conditioning trials.

Although certain patients have such experiences during treatment, not all develop problematic fear and avoidance behaviors. Davey (1989) [18] analyzed this fact and proposed the “latent inhibition hypothesis” to explain it. This author found that 60% of subjects who showed no anxiety before dental treatment did report they had suffered traumatic experiences related to such treatment at some point in their lives. The key to this phenomenon seemed to be that individuals without dental fear suffered their first traumatic experience significantly later than did subjects who presented dental fear. Furthermore, for patients who had not developed dental fear, their traumatic experience occurred after a previous history of favorable or non-averse dental experiences. The previous positive or neutral dental experiences act as a kind of “vaccine” against dental fear in case the subject had an aversive dental experience. Otherwise, as expressed by Davey, a phenomenon of “latent inhibition” would occur. This phenomenon involves a delay or interference in the conditioning of a stimulus because of the subject’s previous exposure to that stimulus.

Some investigations’ results seem to agree. Thus, in the study proposed by Skaret E et al. [19] based in 23-year-old patients who receive several dental treatment sessions with sufficient pain control during childhood and adolescence have an enhanced ability to cope. In addition, they have more confidence in the dentist and a greater degree of satisfaction with dental care.

On the other hand, habituation and awareness processes also affect the dental fear response. Children who had more treatment sessions may become accustomed to the fear more easily; that is, repeated exposure to certain potentially anxious stimuli would make them lose their ability to elicit an emotional response of fear. Patients with few treatment sessions, however, experienced a sensitizing effect to dental stimuli because these patients (who had intensely adverse experiences) are exposed to such stimuli for short, intermittent time periods, when they come irregularly to the dental office. This sensitizing effect could also be developed with treatment sessions, as described by Eysenck [20].

Therefore, previous literature establishes that the classical conditioning mechanism is involved in the origin of children’s dental fear, but traumatic dental experiences alone would be insufficient for a person to develop anxiety about treatments. Other variables, such as patients’ previous history of attending dental consultations, seem to facilitate or hinder the conditioning processes. In this sense, our study aims to identify whether the patients’ age at which visits to the paediatric dentist begin as well as the periodicity of these visits are factors that can prevent dental fear.

## 2. Materials and Methods

### 2.1. Design and Participants

This transversal observational study comprised 545 schoolchildren (49.5% girls and 50.5% boys) aged between 6 and 8 years (M = 6.85, SD = 0.78) who were randomly recruited from 3 primary schools in the Autonomous Community of Madrid (Spain) and whose parents were informed about the study’s objectives prior to the questionnaire.

### 2.2. Procedures and Measures

Each parent/guardian in the household was asked to complete a questionnaire (see online Appendix A). The questionnaires were distributed via classroom teachers and self-administered at home by the participants. Clear instructions were given to avoid confusion, and a researcher’s telephone number was provided to answer questions. All questionnaires were completed anonymously. The questionnaires were previously tested on 10 families to ensure relevance and clarity. Completing the questionnaire took approximately 15 min.

In Spain, preventive dental programs focus on children 6 years old and up. Therefore, the sample was selected from that age. The inclusion criteria were that the children should be between 6 and 8 years old, the parents should have accepted and signed the informed consent, and their mother tongue should be Spanish. The exclusion criteria were uncooperative children do not allow the examination, children with systemic diseases and/or pharmacological treatment.

Ethical approval of this study was obtained from the Ethics Commission for Research of Rey Juan Carlos University (Protocol Code 1604202010520).

The first questionnaire, completed by the parents, collected the family’s sociodemographic information (the child’s age and sex and the parents’ socioeconomic and educational status) and other questions regarding the child’s dental visits, including at what age the child first visited the dentist. It also asked about the frequency of dental visits via a three-point Likert-type scale: 1 = “never gone”, 2 = “when there is a problem or pain”, 3 = “every two or three years”, 4 = “once a year”, and 5 = “every six months”. To evaluate the hypothesis about the latent inhibition phenomenon, we asked whether children had suffered any bad experience at the dentist. The response format was carried out via a dichotomous question (yes/no). If the answer was yes, participants were asked at what age, how many times he had visited the dental clinic before that bad experience, and, finally, if the child was more nervous after that negative visit when going to the dentist. This was registered on a three-point Likert scale: 1 = “no, nothing”, 2 = “a little more”, 3 = “much more” and 4 = “he has not been to the dentist since”.

The second questionnaire was the index of dental anxiety and fear (IDAF-4C) [21], adapted and validated into Spanish by Carrillo-Diaz et al. [22]. The Spanish adaptation shows adequate validity and internal consistency for the Spanish children. This scale measures the four components of dental anxiety (cognitive, physiological, behavioral, and emotional) via 8 items with a Likert-type response format of 5 values, with 1 as the minimum score and 5 as the maximum for each item. Therefore, a higher score indicates greater dental anxiety. The overall score is calculated by adding up the scores, with a range of 8 to 40 points. According to rules established by Armfield [21], the cut-off scores to classify the subject within dental anxiety categories are as follows: no anxiety to mild anxiety from 8 to 15 points, moderate anxiety from 16 to 23 points, and severe anxiety or dental phobia from 24 points and higher. The questionnaire in our sample yielded a Cronbach’s alpha of 0.969, indicating the questionnaire was internally consistent.

### 2.3. Statistical Analysis

Data were analyzed using SPSS V.24.0 (IBM SPSS version 26 Statistics for Windows, Version 24.0, Armonk, NY, USA, IBM Corp). The data analysis included descriptive statistics and the Kolmogorov–Smirnov test to evaluate the assumption of normality, which was confirmed (*p* < 0.05). First, the relationships between variables were analyzed using Pearson’s correlations. Second, a regression analysis determined which factors are predictors of dental fear. Significant levels were established at 0.05.

## 3. Results

A total of 56.70% of the infants evaluated were girls between 2 and 4 years old. Regarding the type of cohabitation, 90.5% of the participants live with both parents, 2.9% live only with the father, 5.7% live only with the mother, and 0.9% live with other relatives.

Firstly, we calculated the correlation between dental fear and the age at which the first visit was made (r = −0.36 *p* < 0.01) and between dental fear and the frequency of visits (r = −0.65 *p* < 0.01). Both were statistically significant as predicted.

To know whether the age at which the patient first went to the dentist and the frequency of visits are factors that could prevent dental fear, a stepwise regression analysis was performed using both factors as independent variables and dental fear as a criterion. This analysis showed that both factors significantly predicted 44.4% of children’s dental fear (Table 1).

On the other hand, to know if the number of positive/neutral/non-averse visits children experienced before a bad experience could prevent or inhibit dental fear, this variable was calculated using the participants in the study who had had a bad experience (*n* = 177). The correlation between this variable and dental fear was significant (r = −0.48 *p* < 0.01), so a greater number of non-averse visits prior to a bad experience lowered the dental fear children developed, despite having that bad experience.

A simple linear regression analysis showed that the number of non-averse visits prior to a bad experience significantly predicts 22.7% of the dental fear in children who have had a bad experience (Table 2).

## 4. Discussion

Our results reveal that the age at which children first go to the dentist and the frequency of visits are key factors in preventing dental fear because they predict 44.4% of dental fear. The applicability of habituation mechanisms can justify these results. If children become familiar with regular dental visits from an early age, they are less likely to develop dental fear [23]. The American Academy of Paediatric Dentistry (AAPD) and the American Dental Association (ADA) recommend that this first visit coincide with the first tooth’s eruption or before the first birthday. If there is no previous pathology/symptomatology, the possibility of carrying out adequate dental education and promotion increases, thus avoiding invasive dental procedures that are very difficult at these ages due to patients’ lack of cooperation [24,25]. Unfortunately, most programs that promote oral health do not consider dental fear, and this consideration which is key to successful dental treatment. Dental fear is associated with a lower use of dental services [25] and with oral problems, such as carious lesions, tooth loss, and the need for oral rehabilitation [26,27].

The frequency of dental experiences seems to act as a positive factor in reducing children’s anxiety. Specifically, our study shows a significant difference when check-ups are performed every 6 months or every year. Thus, visits to the dentist should not be motivated by urgent treatment needs, such as pain, trauma, or caries, as reported by Nicholas et al. [28]. However, a high percentage of the subjects involved in this study (25.7%) only visited the dentist when they already had dental problems or felt pain. In this context, Grembowski and Milgrom [29] reported that the children who were most anxious visited the dentist less frequently or had no previous dental experience.

Our research also contributes to understanding the phenomenon of latent inhibition described in the previous literature. Dental patients who report a history of positive dental experiences are less likely to develop anxiety disorders, despite occasional exposure to painful or unpleasant experiences. In our study, positive or neutral previous visits with negative experiences can predict 22.7% of fear by themselves; therefore, they are a great predictor of dental fear.

It is important to recognize this study’s limitations. First, we used a convenience sample that came from a specific population of children in the community of Madrid, which could limit the possibility of extrapolating results. A possible second limitation comes from using self-report measures, which may be affected by memory and response biases based on social desirability. The children’s recall of their past dental experiences may be incomplete or inaccurate. Third, the study’s cross-sectional design means we cannot adequately capture the nature of the possible dynamic relationships between the sequence of children’s dental visits, the type of adverse experience, and the evolution of the dental fear response. Despite this, the results confirm the hypotheses from previous literature involving this triad, such as the hypothesis of latent inhibition. The results align with previous studies. Finally, not all the etiological factors of dental fear have been measured.

Clinicians should be concerned about how to manage a dental phobic patient because the success of the dental treatment depends on it.

The dentist should try to understand in the first place the patient odontophobia. To overcome this problem dentist should be friendly and try to make the patient to feel comfortable. Even its possible to use audiovisual devices or virtual reality devices [30,31,32].

Eventually, in some extreme cases, if the patient is uncooperative, sedation could be a solution [33].

Regarding the interventions, several contributions have been developed in recent years. The treatment of dental fear comprises the use of drugs such as benzodiazepines [34], relaxation training or behavioral procedures such as stress inoculation or systemic desensitization [35].

The cognitive perspective offers a complementary way of action whose efficacy has been already evaluated [36]. These interventions are focused, on one hand, on identifying the potential cognitive causes of dental fear and, on the other hand, on providing the patients with different tools to control anxiety symptoms.

This paper has relevant implications for nursing practice. To reduce children’s dental fear, efforts should be directed at education. All of this could be successfully achieved via the nursing service because nurses have the most contact with families in the first months of an infant’s life and are most supportive in raising the baby. Therefore, they would act as a link to the child’s dental care. Nurses must inform parents about two fundamental points: the age of initiation into the paediatric dentistry consultation and the importance of periodic revisions. These two aspects help to correctly maintain oral health and to decrease dental anxiety levels. The research on dental fear requires an interdisciplinary approach from researchers and from professionals, among which the work of nursing experts stands out due to its accessibility for data collection and observation of the baby and environment.

Future research with a longitudinal design must elucidate whether our results in the child population stay valid in adolescence and adult life.

## 5. Conclusions

The study helps to clarify that the age of initiation to the paediatric dentist (before 2 years) and the periodic revisions (every 6 months or every year) could protect the child from dental fear.

## Figures and Tables

**Table 1 children-08-01167-t001:** Results of the regression analysis to predict dental fear from the age of the first visit and the frequency of the visit.

Model		Change in R2	β	t
1		0.419 **		
	Frequency of visits		−0.647 **	190.78 **
2		0.028 **		
	Frequency of visits		−0.593 **	170.61 **
	Age at first visit		0.175 **	50.20 **
	R2 Total	0.446 **		
	R2 adjusted	0.444 **		

** *p* < 0.01.

**Table 2 children-08-01167-t002:** Results of the regression analysis to predict dental fear from the number of non-aversive visits prior to a bad experience.

Model		R2	β	t
1		0.232 **		
	Number of non-aversive visits		−0.482 **	−70.27 *
	R2 adjusted	0.227 **		

* *p* < 0.05; ** *p* < 0.01.

## Data Availability

The data that support the findings of this study are available on request from the corresponding author. The data are not publicly available due to privacy and ethical restrictions.

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
