# Peer review of "How Can We Reduce Dental Fear in Children? The Importance of the First Dental Visit"

_children, 2021, doi:10.3390/children8121167_

Round 1
Reviewer 1 Report
Dear Authors,
The article: 'How can we reduce dental fear in children? The importance of the first dental visit' was to identify whether the patients’ age at which visits to the paediatric dentist begin as well as the periodicity of these visits are factors that can prevent dental fear.
English language and style are fine.
The manuscript should be prepared using MDPI guidelines (paragraph indentation, font size, font style, etc.).
Punctuation mistakes should be corrected.
p value must be written in italics.
Add the approval number to the bioethical commission in the materials and methods.
In the statistical analysis, add the names of the statistical tests and the significance level of the tests.
Add a table with abbreviations before references.
Add your questionnaire in the supporting material.
To sum up, article should be reconsider after major revision.
Author Response
The article: 'How can we reduce dental fear in children? The importance of the first dental visit' was to identify whether the patients’ age at which visits to the paediatric dentist begin as well as the periodicity of these visits are factors that can prevent dental fear.
English language and style are fine.
The manuscript should be prepared using MDPI guidelines (paragraph indentation, font size, font style, etc.).
It has been reviewed and modified
Punctuation mistakes should be corrected.
It has been reviewed and corrected
p value must be written in italics.
It has been reviewed and modified
Add the approval number to the bioethical commission in the materials and methods.
Thank you for your appreciation. This information has been added to materials and methods section.
In the statistical analysis, add the names of the statistical tests and the significance level of the tests.
It has been reviewed and added
Add a table with abbreviations before references.
This table has been added
Add your questionnaire in the supporting material.
The questionnaire has been added
Reviewer 2 Report
Dear Authors.
An interesting work.
The itroduction is very good but when mentioning the studies you do not talk about the age range of the studies mentioned. Please put the ages so that the reader can better situate himself.
It would also be interesting to know sociodemographic data in which appears more odontophobia.
Therefore, it would also be interesting to extend the bibliography and add meta-analyses that have been done in different countries on this subject.
It would also be interesting to add information on programs that are provided to cope with dental anxiety. Also proposals.
The discussion is very basic, you should expand more.
Thanks
Author Response
An interesting work.
The introduction is very good but when mentioning the studies you do not talk about the age range of the studies mentioned. Please put the ages so that the reader can better situate himself.
Thank you for your appreciation. This information has been added in the introduction section.
It would also be interesting to know sociodemographic data in which appears more odontophobia.
This information has been added in the introduction section.
Therefore, it would also be interesting to extend the bibliography and add meta-analyses that have been done in different countries on this subject.
Some meta-analyses have been added and the bibliography has been added.
It would also be interesting to add information on programs that are provided to cope with dental anxiety. Also proposals.
It has been reviewed and added in discussion section.
The discussion is very basic, you should expand more.
It has been added more information.